# Association between Leptin (G2548A) and Leptin Receptor (Q223R) Polymorphisms with Plasma Leptin, BMI, Stress, Sleep and Eating Patterns among the Multiethnic Young Malaysian Adult Population from a Healthcare University

**DOI:** 10.3390/ijerph19148862

**Published:** 2022-07-21

**Authors:** Jaiprakash Mohanraj, Urban J. A. D’Souza, Siat Yee Fong, Ivan Rolland Karkada, Heethal Jaiprakash

**Affiliations:** 1Department of Biochemistry, School of Medicine, International Medical University, Kuala Lumpur 57000, Malaysia; jaiprakash@imu.edu.my or; 2Faculty of Medicine & Health Sciences, Universiti Malaysia Sabah, Kota Kinabalu 88400, Malaysia; 3Department of Physiology, Father Muller College of Allied Health Sciences, Father Muller Medical College, Mangalore 575002, India; urbandsouza@fathermuller.in; 4Department of Physiology, Faculty of Medicine, MAHSA Universiti, Jenjarom 42610, Malaysia; ivanrolandk@gmail.com; 5Department of Pharmacology, School of Medicine, International Medical University, Kuala Lumpur 57000, Malaysia; heethaljp@gmail.com

**Keywords:** leptin, leptin receptor, polymorphism, BMI, sleep pattern, eating behaviour, stress, ethnicity

## Abstract

Relative leptin resistance in childhood to absolute leptin resistance in maturity suggests sleep, eating behaviour, and the psychological state as probable causes. The current body of research provides inconclusive evidence linking G2548A and Q223R to obesity. Furthermore, we could find very little data that have observed the association between the environment and gene polymorphism, especially in the multiethnic population that exists in Malaysia. This study searched for a possible link between sleeping habits, eating behaviour, and stress indicators with plasma leptin and its genetic variation in young adult Malaysian healthcare students. The study involved 185 first- and second-year medical and dental students from a healthcare university. Polymerase Chain Reaction–Restriction Fragment Length Polymorphism(PCR-RFLP) determined the genotype, Enzyme Linked Immunoabsorbant Assay (ELISA) tested the serum leptin, and a self-administered questionnaire evaluated sleep, eating behaviour, and psychological condition. Gender and ethnicity are linked to fasting plasma leptin levels (p < 0.001). Plasma leptin also affects stress, anxiety, and sadness. Leptin (*LEP*) and Leptin Receptor (*LEPR*) polymorphisms were not associated with BMI, plasma leptin, sleep, eating behaviour, or psychological state. Young adult Malaysian Indians were obese and overweight, while Chinese were underweight. These findings imply overweight and obese participants were in stage I of leptin resistance and lifestyle change or leptin therapy could prevent them from becoming cripplingly obese as they age.

## 1. Introduction

In the past few decades, metabolic disorders have become a global public health concern [1,2,3]. t has also been witnessed that across the globe, the rate of the adult overweight and obese population grew from 30% in 1980 to 37% in 2013 [4]. Southeast Asia, specifically, has seen a steady rise in overweight and obese individuals (16% to 21% respectively) between 1980 and 2013 [5,6]. According to Malaysian national studies, the prevalence of overweight has increased from 26.7% in 2003 to 29.4% in 2011. The prevalence of obesity has increased from 12.2% in 2003 to 15.1% in 20119 [7]. The second and third National Health and Morbidity Surveys in 1996 and 2006, respectively, reported a three-fold increase in obesity prevalence among adults, surging from 4.4 to 14% over ten years [8]. The rapid increase in this prevalence in developing countries such as Malaysia indicates the role of social, environmental, and behavioural transition [9].

On the other end of the spectrum, Alhazmi showed that there is also a rise in the prevalence of underweight, especially among the young adult female population due to increased urbanisation/globalisation and peer pressure that has compelled this population towards adopting unhealthy practices in weight reduction, thereby increasing the prevalence of underweight, often leading to anorexia, [10,11,12,13].

Many earlier studies identified this double burden on the energy metabolism in childhood as a problem associated with many affluent societies, particularly in countries where children consume unhealthy foods and have poor lifestyle choices [11,14,15,16]. This double burden of malnourishment in the Asian context has been defined, with BMI cut-off levels being modified to ≥23 kg/m^2^, indicating a mild increase in risk. In comparison, a BMI of ≥27 kg/m^2^ marks a high risk for cardiovascular diseases and diabetes [17] and a BMI of ≤18 kg/m^2^ is underweight. Using these diagnostic criteria in Malaysia, at least five large-scale surveys have documented the prevalence of obesity, including a series of National Health and Morbidity Surveys (NHMS). Five Malaysian surveys report one of the highest rates of obesity among the Southeast Asian countries. The most recent estimates published in the NHMS (2015) placed the prevalence of overweight individuals at 30.0% and obesity at 17.0%, a four-time increase in prevalence rates of obesity since 2004 [18]. We were curious to investigate the influences that can contribute to this double burden among the young adult population and inspect for factors that are associated with transforming our prospective robust young adults to bear the brunt of underweight, overweight, and obesity in their adulthood.

Obesity and anorexia are usually identified as disorders disrupting the body fat mass, resulting from impaired energy metabolism [19]. Leptin, a hormone produced by fat cells, is the crucial regulating factor in energy metabolism along with a myriad of other physiological functions [20]. Leptin, released from the body fat store, acts on the satiety centres in the hypothalamus regulating energy metabolism [21]. As shown in Figure 1, leptin binds to leptin receptor primarily in the hypothalamus and through various signalling pathways expresses neuropeptides that regulate energy balance. The discovery of leptin and leptin receptor (LepR) gene products rendered the understanding of the pathognomic pathway in the onset of dysfunction in energy metabolism [20]. Genetic disruption of the *LEPR*s was observed in inherited obesity [22], and it appears to be the most likely reason for the development of human obesity [23]. Various studies associated the dysfunction in energy metabolism with *LEP* and *LEPR* polymorphism [24,25,26]. Dysfunction in energy metabolism in humans was also observed as a pooled effect of genes, environment, lifestyle, and their interaction [27,28].

Leptin, a class I cytokine, primarily affects the hypothalamus, where it binds to the leptin receptor. Activation of the long-form of ObRb, directly and indirectly, activates multiple signalling pathways that involve kinase-induced phosphorylation of proteins, including JAK2/STAT3, erbB2, ERK, IRS1, and rho/rac [30,31]. Signalling requires the presence of intact intracellular domains of the receptor [24]. Most of the studies indicated the preservation of the ObRb isoform of the receptor that has conserved intracellular tyrosine kinase residue which in turn is capable of activating the transcription factor STAT3 [31,32]. Once leptin has bound to its receptor in the arcuate nuclease in the hypothalamus, the expression of neuropeptides that stimulate food intake is suppressed via orexigenic and anorectic neurons [33]. One of the genes activated by leptin-induced STAT3 signalling is SOCS3 (Figure 1), which presents a negative feedback action on the leptin receptor [34]. The amount of leptin production depends on the amount of *LEP* mRNA, which is significantly elevated in isolated adipocytes from obese individuals [35]. It was also reported that insulin may regulate the translation of leptin [36,37]. These findings suggest that regulation of leptin synthesis is primarily at the transcriptional level. The incidence of inactivating mutations of LEP and LEPR is sporadic in humans. It has led to the hypothesis that obesity is a state of “leptin resistance” in which an excess of serum leptin is present. Still, the body does not adequately respond to these increased levels by reducing food intake or body weight [38]. A study conducted on rhesus monkeys concluded that leptin’s role in inhibiting hypothalamic-pituitary-adrenal activation in response to stress [39] is mediated by inhibition of corticotropin-releasing hormone (CRH) synthesis. Thus, we hypothesised that cortisol is a potent stimulus for leptin secretion. On the other hand, activation of the sympathetic nervous system was postulated to be a negative feedback loop to inhibit the synthesis and release of leptin from adipocytes.

In humans, mutations in the coding sequence or at splice sites of LEP cause severe early-onset obesity due to the leptin protein’s inability to signal via its receptor [40,41,42]. However, such mutation is usually sporadic [43]. Several polymorphisms in humans that are attributed to an obesity phenotype [22,44] have been identified in the LEP and LEPR genes: an A to G nucleotide change at position 19 in the 5 V-untranslated region [45]. and a G to A substitution at nucleotide −2548 upstream of the ATG start site [46], both in the LEP gene and in the LEP receptor gene. A single nucleotide change (G to A) in exon 6 of LEPR causes a deficiency in the receptor’s intracellular and transmembrane domains. In a Malaysian study, the occurrence of the LEPR Q223R and K109R alleles were evaluated, and the latter was found to differ significantly by gender [8].

Consequently, we were interested in investigating the association of plasma leptin, *LEP*, and *LEPR* polymorphism in response to the lifestyle-associated factors such as eating behaviour, sleep, and stress experiences among the young adult Malaysian population. We chose this particular population to standardise the effect of type and level of stressors and lifestyle regime that may probably influence the outcome of this research. This study explored a possible association between sleeping habits, eating behaviour, and stress indicators with plasma leptin and its genomic polymorphism among different races in a young adult healthcare student population from Malaysia.

## 2. Materials and Methods

### 2.1. Respondents

This institutional-based cross-sectional case-controlled study was conducted between January to June 2019 among medical and dental students from a Malaysian University. This study was approved by the Research Management Committee, MAHSA University (RMC/E87/2019). In this study, a total of 388 unrelated medical and dental students were included. Previous studies [47,48] used approximately the same sample-size for a similar study. Cohorts involved in this study were briefed on the purpose and benefits of this study, and written consent was obtained only from volunteering students.

In the first phase, 388 students were recruited through the census. A self-administered questionnaire was administered, following which their anthropometric measurements were recorded using a Karada scanner. In the second phase, out of these 388 participants, a stratified random sampling method was adopted to select 185 students based on the inclusion and exclusion criteria.

Inclusion criteria: Malaysian citizens aged ranging between 18 years and 24 years who have not participated in any other clinical study and were willing to sign the written consent form were included in this study. For Group 1, participants were required to have a BMI: Underweight/Overweight/Obese; PSQI: a total score of “5” or greater is indicative of poor sleep quality; DASS: a total score of “20” or greater is indicative of severe to extremely severe stress; and TFEQ: Cognitive Restraint: a total score of “50” or higher is indicative of moderate to bad restraint; Uncontrolled Eating: a total score of “50” or less is indicative of moderate to bad control; Emotional Eating: a total score of “50” or higher is indicative of moderate to poor restraint. They were also required to have plasma leptin of >3200 pg/mL, plasma cortisol of >20 μg/dL, and plasma hsCRP of >3 μg/dL. For Group 2, the participants included had normal BMI, PSQI Score, DASS Score, TEFQ Score, leptin, cortisol, and hsCRP levels

Exclusion criteria: participants suffering from any acute or chronic illness, on any specific diet regime, on any form of treatment, with any disability or deformity, smoker or ex-smoker who stopped smoking for less than six months, or with a drug addiction history were excluded. During the blood sample collection, participants whose blood samples were insufficient, with illness at the time of blood collection, were on medication and/or were unwilling to consent to draw blood were excluded.

All the participants were specially screened for diabetes mellitus, and all had a fasting glucose of less than 110 mg/dL [49]. All the participants in this phase were subjected to estimates of serum leptin, cortisol, and high, sensitive C-reactive protein in plasma. In the third and final phase, out of these 185 students, 62 students were further selected, based on the questionnaire results, BMI (using underweight, overweight, and obese participants as per the WHO guidelines) and plasma leptin, cortisol, and hsCRP. These 62 students were categorised into two groups: case and control. Those included in the case category were selected based on three criteria: Group 1: overweight/obese; Group 2: underweight; and Group 3: with a healthy weight but abnormal sleep, eating, and stress patterns. Those in the control group had healthy BMI with normal sleep, eating, and stress patterns.

### 2.2. Questionnaire

DASS 21 Questionnaire: Depression Anxiety Stress Scale (abbreviated as DASS 21) is a 21 self-administered questionnaire rated on a four-point Likert scale measuring the frequency of participants’ experiences over the last week’s duration. This scale was developed by Lovibond et al. [50] and the validation and reliability of this instrument has been established by various researchers [51,52] and has been commonly used to grade the severity of symptoms of anxiety and depression in both clinical and academic settings. [51,53]. As stated by the content creator, the use of this questionnaire is permissible for all academic purposes.

Pittsburgh Sleep Quality Index (PSQI): This self-administered questionnaire was used to evaluate sleep quality over the past month. The questionnaire contained 19 items used to produce seven components scored on a three-point Likert scale and, finally, one global score. The questionnaire was developed by the University of Pittsburgh [54] for clinical use and research purpose. This has been used to diagnose sleep disorders with excellent efficiency. The validation and reliability of this instrument has been assessed by various researchers [55,56]. The use of this questionnaire has been found in both clinical practice and academic environments [56]. As stated by the content creator, the use of this questionnaire is permissible for all educational purposes.

Three-Factor Eating Questionnaire: The “Three-Factor Eating Questionnaire” (abbreviated as TFEQ) is a questionnaire often applied in food intake-behaviour-related research. It was initially published in 1985 by Stunkard and colleagues [57]. The original TFEQ contains 51 items (questions) and was further revised to TFEQ-R18, referring to current dietary practice. It assesses three different aspects of eating behaviour: restrained eating (conscious restriction of food intake to control body weight or to promote weight loss), uncontrolled eating (tendency to eat more than usual due to a loss of control over consumption accompanied by subjective feelings of hunger), and emotional eating (inability to resist emotional cues). This questionnaire consists of 18 items, out of which 9 concern uncontrolled eating, 6 restrained eating, and 3, emotional eating. This self-administered questionnaire was scored on a four-point Likert scale. Various researchers have well-documented the construct and content validity [58,59]. As stated by the content creator, the use of this questionnaire is permissible for all academic purposes.

A pilot study with 30 participants was conducted before the initiation of this project to test the validity and reliability of all three questionnaires among the current study population. Verbal comments concerning understanding the questionnaire’s content were obtained to assess face validity. All the participants reported ease in completing the survey, while no clarifications were needed. A Cronbach alpha score of >0.75 was observed for all the questionnaires indicating good reliability of these tools.

### 2.3. Anthropometric Measurements

A Karada scanner (Omron, Japan) was used to record the anthropometric mismeasurements of the participants. The Karada scanner is a full-body sensor, body composition monitor, and scale that measures body resistance by using weak currency flowing through both hands and feet. This instrument estimates the body fat percentage by the bioelectrical impedance (BI) method. Muscles, blood, bones, and body tissues with high water content conduct electricity efficiently. On the other hand, body fat does not store much water, and therefore has slight electric conductivity. The Karada scanner algorithm focuses on the bioelectrical impedance method as well as height, weight, age, and gender. The Karada scanner provides an output in the form BMI, total body fat, visceral fat, skeletal muscle distribution and subcutaneous fat (regional distribution), resting metabolic rate, and body age.

### 2.4. Protein Analysis

#### Serum Leptin, Cortisol and hsCRP

Fasting blood samples were collected between 7:00, and 8:30 a.m., one week before their professional examination (barrier exam). The sample collection was designed to be collected during this period when there is significant academic, social, and performance stress on the participants. Plasma was collected using EDTA as an anticoagulant. The samples were centrifuged for 15 min at 1000× *g* at 2–8 °C within 30 min of blood collection. The supernatant was collected and stored at −80 °C until the assay time. Plasma leptin (Catalogue No: DY398-05, R&D Systems, McKinley Place NE, Minneapolis, MN, USA) and hsCRP (Catalogue No: DY1707, R&D Systems, Minneapolis, MN, USA) were assayed by sandwich ELISA, whereas plasma cortisol (Catalogue No: sc-70875, Elabscience, Houston, TX, USA) was assayed with the competitive ELISA technique. All the ELISA kits were of research grade, with a certificate of analysis verified for each.

### 2.5. Genotyping

#### 2.5.1. DNA Extraction

Genomic DNA was extracted from whole blood using the innuPREP Blood DNA Mini Kit from Analytik Jena, Upland, CA, USA) (Catalogue No: AJG#845-KS-1020050). This kit allows the rapid isolation of genomic DNA from EDTA blood samples. The extraction procedure is based on the lysis of the blood sample followed by the binding of the nucleic acids onto the spin filter membrane. After several washing steps, the nucleic acids are eluted from the membrane using elution buffer. The samples were checked for purity and concentration after the extraction procedure. Samples with good purity and concentration were stored under −20 °C until PCR was performed.

#### 2.5.2. Polymerase Chain Reaction (PCR)

Polymerase chain reaction was performed by the qPCR-RFLP method. The primers (TaqPath™ ProAmp™ Master Mixer, Thermo-Fisher Scientific, Scientific Baltics UBA (Saulėtekio al. 15-1, Vilnius, Lithuania) were used to amplify the regions for G2548A and Q223R polymorphism in leptin and the leptin receptor gene, respectively. TaqMan SNP Genotyping Assay for the leptin receptor (Cat No: ABI-4351379 Assay ID: C_8722581_10, SNP ID: rs1137101) and leptin (Cat No: ABI-4351379, Assay ID: C-1328079-10 SNP ID: rs7799039) along with master mix were obtained from Thermo-Fisher Scientific Baltics, USA. PCR was performed using 12.5 µL of master mix, 1.25 µL of primer, up to 11.25 µL of genomic DNA, and up to 25 µL of volume per reaction. The total volume per reaction was 25 µL. PCR starts with a preread at 60 °C for 30 s followed by initial denaturation by enzyme activation at 95 °C for 5 s, further denaturation at 95 °C for 15 s, annealing and extension of the genome at 60 °C for 60 s. The run finally concluded with a postread at 60 °C for 30 s.

## 3. Statistical Analysis

All the statistical analyses of this study were performed using IBM-SPSS, IBM Corp. Released 2017. IBM SPSS Statistics for Windows, Version 25.0. IBM Corp., Armonk, NY, USA. Based on the normal distribution of the sample, student independent two-tailed *t*-test was used to compare the means. ANOVA and Chi-square were used to study the association between different genotypes of G2548A (*LEP*) and Q223R (*LEPR*) polymorphism with plasma leptin and sleep, eating behaviour, and stress patterns. The probability level of *p* < 0.05 was deemed statistically significant in all calculations.

## 4. Results

This study consisted of 185 participants, of which 87 were controls and 95 were cases. A statistically significant difference between the cases and controls of both genders has been recorded. Similarly, there was a statistically significant association between cases and controls among the various ethnic groups (Table 1).

Among these 185 participants, based on the BMI and plasma leptin levels, 63 participants were included, out of which 33 were cases (mean age of 20.91 ± 2.63), and 30 were controls (mean age of 20.78 ± 1.77).


*Relationship between G2548A LEP and Q223A LepR Polymorphisms with Obesity and Plasma Leptin Levels*


The genotypes and frequencies of alleles of G2548A LEP Q223R LEPR polymorphism cases included abnormal BMI and abnormal sleep, eating, and stress patterns, while controls included healthy individuals.


*Relationship between G2548A Lep and Q223A LepR Polymorphisms with Ethnicity*


No significant association was found between ethnic groups with G2548A LEP (*p* = 0.338) and Q223A LepR (*p* =0.151) polymorphisms, as shown in Table 2.


*Relationship between G2548A LEP and Q223A LEPR Polymorphisms with Obesity and Plasma Leptin Levels*


No significant association was found between the measure of leptin with that of G2548A *LEP* (*p* = 0.196) and Q223A *LEPR* (*p* = 0.453) polymorphisms, respectively. Similarly, no significant correlation was observed between the measure of BMI with that of G2548A *LEP* (*p* = 0.117) and Q223A *LEPR* (*p* = 0.469) polymorphisms, respectively. These results are represented in Table 3 and Table 4.


*Relationship between LEP G2548A and LEPR Q223A Polymorphisms with Obesity and Its Associated Risk Factors*


Similarly, the distribution of *LEP G2548A*, *LEPR Q223A* polymorphisms, and various obesity and its associated risk factors showed no significant association between any of the variables. These results are represented in Table 5.


*Relationship between G2548A LEP and Q223A LEPR Polymorphisms with Group 1 (Overweight to Morbidly Obese) Cases*
*G2548A LEP and Q223A LEP*R polymorphism with Group 2 (Underweight to Anorexic) cases:*G2548A LEP and Q223A LEP*R polymorphisms with Group 3 with normal BMI and abnormal sleeping, eating, and stress factors:

## 5. Discussion

The present study investigated the association between *G2548A* polymorphism for the leptin gene and *Q223R* polymorphism for the LepR gene with fasting plasma leptin and BMI (markers for dysfunction in energy metabolism), with sleep, eating, and stress patterns as risk factors associated with altered body weight among the Malaysian young adults. Several studies that have investigated the association of leptin and LepR with obesity patterns have indicated several limitations with confounding factors, such as environment, psychological state, food habits, etc. [24,25,26]. Thus, in this study, we investigated the most common and well-archived factors such as sleep, eating behaviour, stress, anxiety, and depression affecting the genome involved in energy metabolism.

Leptin, released from body fat stores, acts on the satiety centre in the hypothalamus regulating energy metabolism [21]. The discovery of leptin and the *LEPR* gene have provided a better understanding of the pathognomic pathways in the onset of dysfunction in energy metabolism [20]. Genetic disruptions of the LepR have been observed in inherited obesity [22], and it appears to be the most likely reason for the development of human obesity. On the other end of the spectrum, elevated plasma leptin levels have been known to cause eating disorders leading to underweight and anorexia [60,61]. Thus, leptin, one of the cytokines released in response to infection and inflammation, may directly affect the brain, such as inducing anorexia [62]. Various studies have been associated with the dysfunction in energy metabolism with *LEP* and *LEP*R polymorphisms [63,64,65]. Dysfunction in energy metabolism in humans is often a combined effect of genes, environment, lifestyle, and their interaction [27,28]. Consequently, *LEP* and *LEP*R polymorphism could influence the response to a healthy environment, such as eating behaviour, sleep, and stress experiences, especially among the young adult population [28].

Further, it is also known that both the *G2548A LEP* and *Q223A LEP*R polymorphisms are associated with amino acid substitutions in the extracellular region of the *LEP* and *LEP*R and have a potential functional consequence [64]. Both the polymorphisms resulted in changes in the charge [glycine (G) to alanine (A) at codon 2548 in the leptin gene] and [glutamine(Q) to arginine(R) at codon 223 in the leptin receptor gene], which is most likely to have a functional consequence [66]. Studies have shown alterations in other coding and noncoding sequences of *LEPR* and *LEP*, among which most of these are either silent mutations or very rare [24,66,67,68].

In the present study group, among the 89 cases with abnormal BMI, a highly significant fasting plasma leptin level was recorded in females compared to males (Table 1). This result substantiates earlier work done by Peltz and co-workers on the correlation of leptin with body fatness among Mexican-Americans [69]. This study further suggested that measuring hip and waist circumferences and BMI has the same predictive value on plasma leptin. In the present study, we observed a significant association for *LEP G2548A* in Group 2, especially when SNP was compared between genders. These findings concurred with a similar survey done by Fan and Say [8]. However, it was observed that there was no significance in the *LEPR* gene polymorphism between males and females. Since leptin links the body adiposity with the hypothalamic regulation of reproduction [70,71], these results indicate that the *LEPR* gene has not yet been affected by the influence of sex hormones.

A significant difference in the mean scores was observed among Malay, Chinese, Indian, and Other ethnic populations for fasting plasma leptin when compared between cases and controls. The Chinese community showed significantly (*p* < 0.000) lower levels of leptin while the Indian (*p* and mixed population showed raised levels of leptin, both of these were statistically significant with *p* < 0.000. These findings concurred with the work [6]. The elevated levels of plasma leptin found in a specific population can be attributed to the presence of a high quantity of subcutaneous fat (SCF), especially among Malaysian Indians and mixed race populations [72,73]. The low levels of leptin among Malaysian Chinese can be attributed to the low SCF; further, it was also observed that among the obese Chinese population, there was a high amount of visceral fat when compared to SCF, especially among males. It is a well-established fact that the production of leptin is higher from the SCF when compared to the visceral fat as indicated in earlier reports [74,75,76]. Results of the genotyping study, as shown in Table 2, indicate no significant association in their expression among the three subgroups between the four ethnic populations; this was contradictory to the finding shown by [8]. This contradiction could be attributed to the specific age group (between 18 and 21) selected in this study. Studies have shown that the prevalence in the development of leptin resistance is more common among the older adult population [77,78], while children exhibit absolute leptin resistance due to inheritance of an altered *LEPR* gene or “relative leptin resistance” to support increased growth and development of reproductive capacity [79]. However, the concern remains on this metabolic syndrome’s progress into adulthood.

As indicated in earlier studies [73,80,81], our study also showed the mean values of BMI to be higher among the male and Indian subgroups in the Malaysian population. The BMI of the study population was further subgrouped based on the WHO criteria for obesity. It was observed that there was a statistically significant (*p* < 0.01) association of fasting plasma leptin levels with that of BMI subgroups (Table 1). This indicates a positive association between BMI increase and leptin production from fat cells. Van Rossum and colleagues reported that the individuals with higher leptin levels had a higher risk of gaining weight, showing that elevated leptin production was directly related to the mass of subcutaneous fat cells in the body [82]. Leptin, a suppressor of non-satiety, is actively produced to reduce food intake and thereby regulate energy balance [83].

On the other hand, anorexia–cachexia syndrome appears to be a multifactorial disorder [84] attributed to a direct link with leptin as a class I cytokine involved in its pathobiology. Altered leptin levels in obesity and anorexia have also been associated with leptin and leptin receptor gene polymorphism. Contradictory to these views, in our study population, as shown in Table 2, there appears to be no significant association between the three subgroups with *LEP* and *LEP*R gene polymorphism. However, clinical (BMI) signs that corroborated with plasma leptin for underweight and obesity exist, and there appear to be no significant polymorphic changes observed in the *LEP* and *LEP*R gene.

Various pathophysiological, social, psychological, and environmental factors contribute to energy metabolism disorder [85]. Hence, we investigated the well-documented contributors to obesity, such as sleeping and eating behaviours and stress, anxiety, and depression patterns among the study populations. As shown in Table 1, our results indicate a statistically significant (*p* < 0.001) association between stress, depression, and anxiety with plasma leptin, whereas sleep and eating behaviour had no association. The student blood samples to assess plasma leptin were collected one week before the start of their professional examination (barrier exam). This was intended to ensure that the nature of the stressors involved in the energy metabolism pathway was standardised. The results from our study indicate that those subjects who experience high stress, anxiety, and depression have their plasma leptin elevated. A similar study done by [86,87,88] showed that elevation of leptin in response to psychological stress could be due to increased production of cortisol which in turn increases the production of leptin from the adipocytes.

Leptin gene polymorphism was studied by assessing *G2548A* in the study population. It is known that glycine mutation to alanine at codon 2548 alters the promoter site involved in the production of leptin [89]. Our results indicate no significant association between plasma leptin with LEP G2548A in our study population. This suggests that the elevated levels of plasma leptin observed among the obese and the low levels of leptin in the underweight population primarily depended on the mass of fat tissue [90]. The fact that no significant polymorphism has been observed indicates that these young adults are in stage 1/Early stage [77] of dysfunctional energy metabolism and thus are within the scope of implementing lifestyle modification or administration of LRT to correct this alteration. The study of *LEPR* gene polymorphism showed no association with plasma leptin.

In numerous populations and age groups, the effect of *LEP* and *LEPR* polymorphisms on the risk of obesity has been well-documented. None of the SNP genotype and allele frequencies were associated between Groups 1, 2, or 3. (Table 6, Table 7 and Table 8). They were also unrelated to gender and race within these groups. Similar results were found in studies conducted by Fan & Say, although they reported a significant association among ethnic groups [8]. This may be due to the fact that, in contrast to their study, our study population matched in terms of age, gender, and ethnicity across all groups. In line with previous research, this study was unable to establish a link between G2548A and obesity. However, research among Tunisians, Pacific Islanders, Spaniards, and Africans supports an association between G2548A and underweight. The *LEP* SNP (rs7799039) is a transition from glycine to alanine at position −2548 upstream of the ATG start site in the LEP gene5’ promoter gene [23]. Jiang observed in his paper that the *LEP G2549A* polymorphism was not located in a conserved region in humans, despite popular belief. In contrast, Hinuy and colleagues reported that the polymorphism in the promoter region of adipocytes may contain transcription-inhibiting elements. Fan and co-workers [8] concluded that the presence of *G2548A* polymorphism on leptin expression must be investigated further. Our research supports these claims, suggesting that the *G2548A* polymorphism is likely nonfunctional.

The *LEPR Q223R* SNP(rs1137101) is an A to G transition that causes a nonconservative change that converts glutamine(Gln/Q) to arginine(Arg/R) on codon 223 (CAG to CGG) at position 668 in exon 6 [24]. This polymorphism results in substituting a neutral amino acid to a favourable amino acid in the region located within the leptin-binding site leading to impaired signalling capacity. Further, *LEP Q223R* polymorphism was evidenced for positive natural selection in major ethnic groups among Asians [91]. Our study’s results agree with earlier studies that reported no association between *LEPR Q223R* with obesity [24,57,92]. However, conflicting reports established the association with obesity [93,94,95]. Our findings also show the lack of association of environmental factors, such as eating behaviours, psychological factors such as anxiety, stress, and depression, and physiological factors such as sleep, with gene polymorphism.

In conclusion, our results demonstrate the difficulty in conclusively explaining human obesity in terms of the well-known *LEP* and *LEPR* genes. Fan and colleagues [8] found that a combination of different allelic variants from different genes can obscure the effect of these genotypes on healthy, obese, or underweight individuals. The lack of association merely demonstrates the complexity of the pathobiological development of obesity, not its absence.

Our results further substantiated the nonfunctional role of investigating *LEP* and *LEPR* genes as causative in a complex energy metabolism disorder. We were also able to recommend that among young individuals in our study population representing both sides of the spectrum in energy metabolism, gene polymorphism was not found to be significant, thus re-emphasising the fact that the pathology exists in the mass of fat cells. Although other similar studies have used approximately the same sample size [91,96], our research had the innate issue of sample size that could have influenced the output. There is always a broad scope of studying the role of another functional gene concerning factors affecting energy metabolism to understand its implication in etiopathogenesis. Confounding factors such as ghrelin and insulin could be considered to further our understanding of the association with SNPs on the various related genes involved in energy metabolism.

Anorexia and obesity are disorders of energy metabolism known to be altered by leptin dysfunction. This is now understood to be mainly due to mutations in LEP and *LEPR* by many authors [97,98]. Enriori et al. [77] showed that leptin resistance develops in three stages:Stage 1 (Early Stage) is when the subject gains weight but maintains adequate response to the anorectic effect of peripheral leptin:Stage 2 (Middle Stage) is when the subjects develop leptin insensitivity due to lack of STAT3, expressed by changes in food intake and body weight but continue to respond to leptin injections:Stage 3 (Final Stage) is when the subjects develop central leptin resistance [99,100,101].

Thus, for a population of young adults who show signs of overweight and obesity, leptin replacement therapy would probably be beneficial [102,103]. Administration of leptin has shown the reversal of neuroendocrine and metabolic abnormalities in an individual with congenital leptin deficiency [102,103] but has also restored regular menstrual cycles, corrected gonadal abnormalities, and improved bone mineral density [104]. Hence, among the youth population, early detection and active intervention of dysfunctions in energy metabolism can reduce morbidity as they reach adulthood.

## 6. Conclusions

It is further recommended to investigate stress, sleep, and eating behaviour in a controlled environment that can standardise the cases before analysing the polymorphic changes. It is also suggested to consider conducting a longitudinal study observing the effect of the parameters mentioned above in the development of these weight-related issues.

## Figures and Tables

**Figure 1 ijerph-19-08862-f001:**
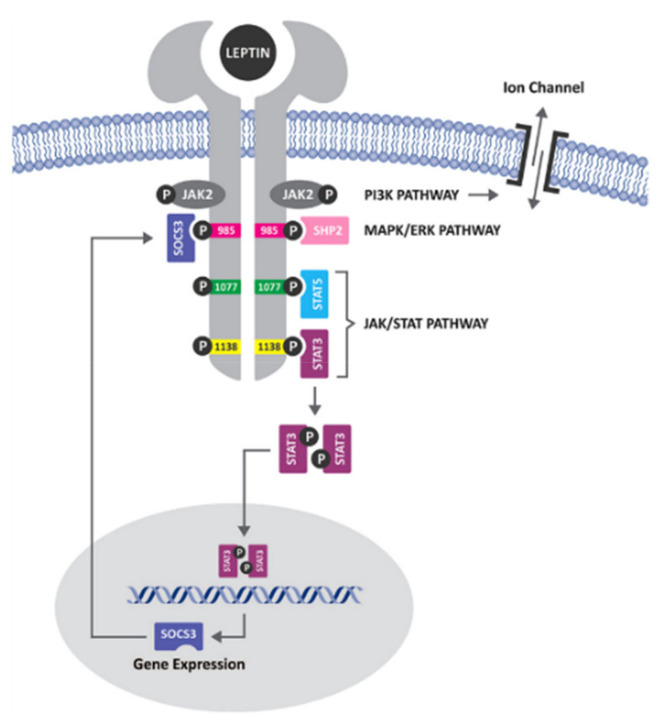
Leptin signalling pathway [29].

**Table 1 ijerph-19-08862-t001:** ANOVA test results of serum leptin (pg/mL) levels between control and cases based on gender, ethnicity, and BMI.

	Gender		Mean	Std. Deviation	F	Sig.
Gender	Male	Cases (25)	2955.13	731.23	22.175	0.000
		Controls (29)	2184.09	458.41		
	Female	Cases (64)	3392.66	378.75	11.527	0.001
		Controls (67)	3009.46	823.43		
Ethnicity	Malay	Cases (23)	3322.19	570.03	0.195	0.660
		Controls (38)	3241.81	750.19		
	Chinese	Cases (21)	3113.42	606.56	10.016	0.003
		Controls (24)	2555.61	574.94		
	Indian	Cases (32)	3294.17	512.29	18.267	0.000
		Controls (24)	2529.84	822.21		
	Others	Cases (13)	3369.45	402.55	45.295	0.000
		Controls (10)	1973.31	592.86		
BMI	Underweight (29)		2880.96	574.43	2.71	0.01
	Normal (100)		2877.77	793.31		
	Overweight (29)		3174.01	726.22		
	Grade 1 Obesity (19)		3321.97	649.65		
	Grade 2 Obesity (3)		3655.30	267.10		
	Grade 3 Obesity (5)		3705.04	221.08		

A significance of *p* < 0.05 was assigned for this analysis. Others under ethnicity include mixed ethnic groups within Malaysia.

**Table 2 ijerph-19-08862-t002:** Descriptives Statistics of leptin, LEP, and LEPR observed in both the gender and ethnic population.

	Gender [Mean (SD)]		Ethnicity [Mean (SD)]			
	Male (*n* = 29)	Female (*n* = 34)	Malay (*n* = 20)	Chinese (*n* = 20)	Indian (*n* = 18)	Others (*n* = 5)
BMI	26.43 (9.73)	23.08 (6.16)	25.71 (5.66)	23.78 (5.86)	26.15 (11.8)	18.12 (5.6)
Leptin (pg/mL)	2568.54 (717.7)	3296.47 (461.3)	3048.26 (786.4)	2814.48 (620.2)	2988.71 (726.0)	3103.28 (504.97)
*LEP G2548A* % (*n*)						
Ala/Ala	48.3 (14)	44.1 (15)	45 (9)	60 (12)	33.3 (6)	40 (2)
Ala/Gly	34.5 (10)	41.2 (14)	30 (6)	40 (8)	44.4 (8)	40 (2)
Gly/Gly	17.2 (5)	14.7 (5)	25 (5)	0 (0)	22.2 (4)	20 (1)
*LEP*R *Q223R* % (*n*)						
Arg/Arg	6.9 (2)	8.8 (3)	0 (0)	10 (2)	11.1 (2)	20 (1)
Arg/Gln	37.9 (11)	35.3 (12)	60 (12)	20 (4)	33.3 (6)	20 (1)
Gln/Gln	55.9 (16)	36.5 (19)	40 (8)	70 (14)	55.6 (10)	60 (3)

A *p*-value of < 0.005 was observed in Group 2 when LEP G2548A was compared between genders.

**Table 3 ijerph-19-08862-t003:** Chi-square test result between LEP and LEPR genotype among different ethnic subpopulations.

Genotype	Ethnicity				*p*-Value (χ^2^)
	Malay	Chinese	Indian	Others	
LEP G2548A					
Ala/Ala	9 (31%)	12 (41.4%)	6 (20.7%)	2 (6.9%)	0.338 (df = 6)
Ala/Gly	6 (7.6%)	8 (33.3%)	8 (33.3%)	2 (8.3%)	
Gly/Gly	5 (50%)	0 (0%)	4 (40%)	1 (10%)	
LEPR Q223R					
Arg/Arg	0 (0%)	2 (40%)	2 (40%)	1 (20%)	0.151 (df = 6)
Arg/Gln	12 (52.2%)	4 (17.4%)	6 (26.1%)	1 (4.3%)	
Gln/Gln	8 (22.9%)	14 (40%)	10 (28.6%)	3 (8.6%)	

A *p*-value of <0.05 is significant.

**Table 4 ijerph-19-08862-t004:** Comparison of the mean leptin and BMI values in different genotypes of G2548A Lep and Q223A LepR polymorphisms in the studied population.

Genotypes	Number	Leptin Conc. ^1^	*p*-Value	BMI ^2^	*p*-Value
*LEP* G2548A					
Ala/Ala	29	2794.19 ± 649.18	0.196 (F = 1.67)	22.46 ± 4.7	0.117 (F = 2.22)
Ala/Gly	24	3074.30 ± 685.07		27.0 ± 10.6	
Gly/Gly	10	3175.28 ± 786.74		27.9 ± 7.9	
*LEPR* Q223R					
Gln/Gln	35	2929.59 ± 692.58	0.453 (0.801)	24.1 ± 6.8	0.469 (F = 0.768)
Arg/Gln	23	3074.55 ± 729.77		26.0 ± 10	
Arg/Arg	5	2663.44 ± 816.27		21.5 ± 6.9	

Data are represented as mean ± SD; ^1^ pg/mL; ^2^ kg/m^2^.

**Table 5 ijerph-19-08862-t005:** ANOVA test result of leptin, *G2548A LEP*, and *Q223A LEP*R polymorphisms with associated risk factors.

Parameter (*n* = 63)	Plasma Leptin *p*-Value (F Score)	*LEP G2548A* *p*-Value (F Score)	*LEPR Q223R* *p*-Value (F Score)
PSQI	0.2170 (1.53)	0.219 (1.55)	0.797 (0.22)
Stress	0.000 (23.89)	0.078 (2.22)	0.959 (0.15)
Depression	0.001 (5.19)	0.492 (0.86)	0.841 (0.35)
Anxiety	0.000 (8.70)	0.279 (1.30)	0.457 (0.92)
TFEQ (Cognitive Restraint)	0.3250 (1.16)	0.703 (0.47)	0.947 (0.12)
TFEQ (Uncontrolled Eating)	0.0200 (3.35)	0.554 (0.59)	0.512 (0.67)
TFEQ (Emotional Eating)	0.0100 (3.73)	0.590 (0.64)	0.529 (0.74)

A *p*-Value of < 0.005 is significant.

**Table 6 ijerph-19-08862-t006:** Results of *G2548A LEP* and *Q223A LEPR* of controls and Group 1 cases with BMI > 29.9 (Overweight to Morbidly Obese) and no abnormal sleeping, eating, and stress parameters.

Genotypes	Controls % (*n*)	Cases % (*n*)	Odds Ratio	95% CI	*p*-Value
*LEP G2548A*					
Ala/Ala	50 (15)	28.6 (4)	1 *		
Ala/Gly	30 (9)	57.1 (8)	0.3	0.06–1.28	0.11
Gly/Gly	20 (6)	14.3 (2)	0.8	0.11–5.50	0.83
Alleles					
Ala-Allele	72.2 (39)	61.5 (16)	0.6	0.22–1.65	0.33
Gly-Allele	27.8 (15)	38.5 (10)			
*LEPR Q223R*					
Gln/Gln	53.3 (16)	57.1 (8)	1 *		
Arg/Gln	40 (12)	42.9 (6)	1	0.27–3.6	1
Arg/Arg	6.7 (2)	0 (0)	2.5	0.11–59.9	0.5
Alleles					
Gln-Allele	75.9 (44)	78.6 (22)	1.6	0.39–3.45	0.78
Arg-Allele	22.2 (14)	21.4 (6)			

* Reference group.

**Table 7 ijerph-19-08862-t007:** Results of *G2548A Lep* and *Q223A LepR* of controls and Group 2 cases with BMI < 18 (Underweight to Anorexic) and no abnormal sleeping, eating, and stress parameters.

Genotypes	Controls % (*n*)	Cases % (*n*)	Odds Ratio	95% CI	*p*-Value
*LEP G2548A*					
Ala/Ala	50 (15)	63.6 (7)	1 *		
Ala/Gly	30 (9)	27.2 (3)	0.3	0.06–1.28	0.11
Gly/Gly	20 (6)	9.1 (1)	0.8	0.11–5.50	0.83
Alleles					
Ala-Allele	72.2 (39)	77.3 (17)	0.6	0.22–1.65	0.33
Gly-Allele	27.8 (15)	22.7 (5)			
*LEPR Q223R*					
Gln/Gln	53.3 (16)	54.5 (6)	1 *		
Arg/Gln	40 (12)	45.5 (5)	1	0.27–3.6	1
Arg/Arg	6.7 (2)	0 (0)	2.5	0.11–59.9	0.5
Alleles					
Gln-Allele	75.9 (44)	68.2 (15)	1.6	0.39–3.45	0.78
Arg-Allele	22.2 (14)	31.8 (14)			

* Reference group.

**Table 8 ijerph-19-08862-t008:** Results of *G2548A LEP* and *Q223A LEPR* of controls and Group 3 cases with normal BMI and abnormal sleeping, eating, and stress parameters.

Genotypes	Controls % (*n*)	Cases % (*n*)	Odds Ratio	95% CI	*p*-Value
*LEP G2548A*					
Ala/Ala	50 (15)	37.5 (3)	1 *		
Ala/Gly	30 (9)	50 (4)	0.3	0.06–1.28	0.11
Gly/Gly	20 (6)	12.5 (1)	0.8	0.11–5.50	0.83
Alleles					
Ala-Allele	72.2 (39)	62.5 (10)	0.6	0.22–1.65	0.33
Gly-Allele	27.8 (15)	37.5 (6)			
*LEPR Q223R*					
Gln/Gln	53.3 (16)	62.5 (5)	1 *		
Arg/Gln	40 (12)	25 (2)	1	0.27–3.6	1
Arg/Arg	6.7 (2)	12.5 (1)	2.5	0.11–59.9	0.5
Alleles					
Gln-Allele	75.9 (44)	75 (12)	1.6	0.39–3.45	0.78
Arg-Allele	22.2 (14)	25 (4)			

* Reference group.

## Data Availability

The data presented in this study are openly available in https://figshare.com/s/032306ca1788d5ac000f.

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
