# Peer review of "Association between Leptin (G2548A) and Leptin Receptor (Q223R) Polymorphisms with Plasma Leptin, BMI, Stress, Sleep and Eating Patterns among the Multiethnic Young Malaysian Adult Population from a Healthcare University"

_ijerph, 2022, doi:10.3390/ijerph19148862_

Round 1
Reviewer 1 Report
In this article, Mohanraj et al study the association of polymorphisms in leptin and leptin receptor genes with a multitude of factors in the Malaysian population. Although the findings are interesting, there are several concerns with limit the enthusiasm in this study:
1. The study attempts to incorporate and associate multiple factors including sleep patterns, stress, diet, etc. It is critical to note that leptin signaling is a complex phenomenon that is regulated by multiple factors. Although these are crucial factors in the context of obesity and metabolism, the way this study is designed makes it inconclusive, which could be false negatives, in most aspects. For example, the authors posit that “sleep and eating behavior had no association”. However, it has been previously established that these factors contribute to alterations in leptin levels. Now, whether the absence of alterations they see is a true finding or is merely due to the issue of sample size/ study design cannot be confidently interpreted. The authors need to rework to simplify their analysis and explain such discrepancies.
2. The authors need to make significant changes to the write-up. The introduction sounds extremely distorted and does not flow well. The authors need to organize it to include relevant background, followed by rationale, the hypothesis (if any), and then start the results. The results must be properly explained. There needs to be a scholarly discussion of the findings and the implications of both positive and negative associations they observed.
3. The statistics in the introduction and discussion sections are considerably older. They need to be updated to the recent ones. Furthermore, the background on the important topics like leptin or leptin receptor gene polymorphisms is missing.
4. The authors have included many bold statements with enough scientific evidence. E.g “We were also able to recommend that among young individual on both sides of the spectrum gene polymorphism is not significant and established the fact that the pathology exists in the mass of fat cells”. They need to either tone it down or delete it completely.
5. The article in several sections is difficult to comprehend as it consists of severe grammatical errors, including spelling mistakes, unclear sentence structures, consistency, spacing, tenses, unnecessary apostrophes, and other punctuations. If accepted, the article needs to undergo thorough proofreading by professionals.
6. The table structure as well as legends are unclear and should be improved.
Author Response
"Please see the attachment.

Reviewer 2 Report
In this manuscript, authors have provided an insight of the correlations between plasma leptin, leptin receptor polymorphisms and BMI, ethnicity, stress. The manuscript is centered in particular in Malaisyan population, and present characters of novelty and interest; however little changes could improuve it overall.
First of all, I suggest going through moderate English revision, some phrases in the manuscript are difficult to understand and need to be read twice to catch their meaning.
The abstract needs to be clearer and to contain a little bit more introductory information and a best resume of results.
In the introduction, it may be opportune to add a figure about leptin signalling, showing all the pathways deriving from leptin receptor activation, and at the same time underline the factors that influence leptin production / activation. This part is well redacted in the introductio, and the presence of a figure will help to understand it more immediately.
In methods, a better description (maybe in a table) of exclusion and inclusion criteria is necessary; as well as a little paragraph justifying the numerosity of the study ( what statistical analysis were run to determine the numerosity of each group?)
In results, it will be best to underline significant results by using bold or other forms; and adding a little paragraph at the end of the tables resuming all the important results achieved. At the same time, it will be necessary to add in that paragraph the details of group1, 2, and 3; that are cited in discussion but not so evident in results.
In discussion, it will be better to change line 328 with "one of the major factors for...", since lots of factors can influence obesity development in addition to leptin resistance. In line 407, it may be necessary to add that the low numerosity could be responsible of the non significant data. Moreover, it will be better to differenciate the stages of leptin resistance with a punctuation list, rather than in a written paragraph (line 489).
Less important minor changes necessary are:
-Obrb line 86, needs to be explicited before
- line 95, a reference appears as #110
-line 256 ..
-line 357, SNP needs to be explicited before, same as SCF line 372 and LRT line 432.
Author Response
Please see the attachment
An extensive grammar check was performed and corrections were made wherever necessary.

Reviewer 3 Report
In this paper, Jaiprakash and colleagues reported the association between leptin (G2548A) and leptin receptor (Q223R) polymorphisms with plasma leptin, BMI, stress, sleep, and eating patterns among the young adult Malaysian population. The study is straightforward, and the manuscript is well-written. The interpretation is consistent with the findings. However, my main concern is the sample size and power of the study. Many parameters are being studied using very less samples. Moreover, the study populations have been further divided into Malay, Chinese, Indian, and others. This further weakens the power of the study. It would have been better if the authors had performed an independent analysis with each ethnic group using more samples. Did the authors check the power of the study? I would highly recommend increasing the sample size. There are two tables named Table 3. Table 5; Association study was conducted by combining all the ethnic groups and therefore, the results obtained, cannot be accepted with confidence.
Also, this manuscript has minor punctuation and grammatical errors in a few places.
I have listed some of them.
Line 16-17>>Current evidences provide inconclusive evidence linking G2548A and Q223R to obesity>> Sentence is not properly constructed, consider revising it. Also, consider changing the word ‘evidences’.
Line 106-107>>A study conducted in rhesus monkey>>> A study conducted on rhesus monkeys.
Line 136>> Malaysian’s between the age group of 18 to 25years>> Malaysians
Line 156-158>> This scale was developed by (Lovibond & Lovibond, 1995), the validated and reliability of this instrument has been>> This sentence is not properly constructed.
Line 161>> As stated by the content creater,>> creator
Line 254>> between different genotype of>> genotypes
Table 1>> Ethinicity>> Ethnicity
Table 2>> Paramenter>> Parameter
Line 328>> On the other end of the spectrum, elevated levels of plasma leptin have known>> have been known
Line 364>> population>>populations
Author Response
All the grammatical corrections suggested were made in the document using track changes.
Please see the attachment for responses to reviewers' comments.

Round 2
Reviewer 1 Report
The authors have tried to address most of my concerns. Having said that, I still think the major issue of sample size still remains unanswered.
Reviewer 3 Report
The authors have addressed some of the concerns. However, they need to increase the sample size and refine the statistical methods. The result section is confusing and the study outcome is not explained well.
